# Antagonistic modulation of NPY/AgRP and POMC neurons in the arcuate nucleus by noradrenalin

Lars Paeger[1,2], Ismene Karakasilioti[2,3,4,5], Janine Altmüller[4,6], Peter Frommolt[7], Jens Brüning[2,3,4,5], Peter Kloppenburg[1,2]*

[1]Biocenter, Institute for Zoology, University of Cologne, Cologne, Germany; [2]Excellence Cluster on Cellular Stress Responses in Aging Associated Diseases, Cologne, Germany; [3]Max Planck Institute for Metabolism Research, Cologne, Germany; [4]Center of Molecular Medicine Cologne, University of Cologne, Cologne, Germany; [5]Center for Endocrinology, Diabetes and Preventive Medicine, University Hospital Cologne, Cologne, Germany; [6]Cologne Center for Genomics, University of Cologne, Cologne, Germany; [7]Bioinformatics Facility, Excellence Cluster on Cellular Stress Responses in Aging Associated Diseases, Cologne, Germany

**Abstract** In the arcuate nucleus of the hypothalamus (ARH) satiety signaling (anorexigenic) pro-opiomelanocortin (POMC)-expressing and hunger signaling (orexigenic) agouti-related peptide (AgRP)-expressing neurons are key components of the neuronal circuits that control food intake and energy homeostasis. Here, we assessed whether the catecholamine noradrenalin directly modulates the activity of these neurons in mice. Perforated patch clamp recordings showed that noradrenalin changes the activity of these functionally antagonistic neurons in opposite ways, increasing the activity of the orexigenic NPY/AgRP neurons and decreasing the activity of the anorexigenic POMC neurons. Cell type-specific transcriptomics and pharmacological experiments revealed that the opposing effect on these neurons is mediated by the activation of excitatory $\alpha_{1A}$ - and $\beta$- adrenergic receptors in NPY/AgRP neurons, while POMC neurons are inhibited via $\alpha_{2A}$ – adrenergic receptors. Thus, the coordinated differential modulation of the key hypothalamic neurons in control of energy homeostasis assigns noradrenalin an important role to promote feeding.

*For correspondence: peter.kloppenburg@uni-koeln.de

## Introduction

Substantial progress has been made identifying central neuronal circuits that are crucial for controlling feeding behavior and energy homeostasis. Understanding the cellular and molecular mechanisms mediating information processing in neuronal circuits that regulate energy homeostasis is essential to devise specific pharmacological strategies, which assist obese patients in starting and maintaining a program of weight loss with no or minimized side effects on other neuronal systems (*Gouni-Berthold et al., 2013*; *Holes-Lewis et al., 2013*; *Misra, 2013*). Important microcircuits in control of energy balance are localized in the arcuate nucleus of the hypothalamus (ARH), which contains two main neuron populations with antagonistic functions: Neuropeptide Y and agouti-related peptide (NPY/AgRP) co-expressing neurons signaling hunger, promoting food intake, and pro-opio-melanocortin (POMC) - expressing neurons which signal satiety (*Sohn et al., 2013*; *Varela and Horvath, 2012*).

Many important fuel-sensing endocrine and metabolic factors, such as insulin, leptin, glucose, serotonin, dopamine, free fatty acids and uridine diphosphate (*Gao and Horvath, 2007*; *Jo et al.,*

*2009*; *Jordan et al., 2010*; *Sohn et al., 2011*; *Steculorum et al., 2015*; *Varela and Horvath, 2012*; *Zhang and van den Pol, 2016*) have been identified and various immediate actions of these modulators and nutrient components on the POMC and AgRP neurons are well understood (*Belgardt et al., 2008*; *Claret et al., 2007*; *Jo et al., 2009*; *Parton et al., 2007*; *Spanswick et al., 1997*, *2000*). However, the effect of the catecholamine noradrenalin (NA) on the ARH is not well defined. Noradrenalin is associated with a number of important CNS functions (*Szabadi, 2013*) including energy homeostasis and food intake (*Commins et al., 1999*; *Ste Marie et al., 2005*; *Thomas and Palmiter, 1997*; *Wellman, 2005*, *2000*). Potentially, NA could be a direct modulator of the energy homeostasis-regulatory POMC- and AgRP neurons in the ARH, since noradrenergic projections from the hindbrain (A1, A2 noradrenergic cell groups) and the locus coeruleus (LC) to the ARH have been described in rats (*Fraley, 2006*; *Fraley and Ritter, 2003*; *Yoon et al., 2013*) and manipulation of the noradrenergic system in the hypothalamus has been found to modify feeding behavior and food intake. Impairment of the dorsal or the ventral noradrenergic bundle, which contain efferent projections from the LC to the forebrain, caused hyper- or hypophagia, respectively (*Ahlskog and Hoebel, 1973*; *Hoebel et al., 1989*). In line with these lesion experiments, local NA injections into defined hypothalamus regions either elicits or inhibits feeding in rodents (*Booth, 1967*; *Leibowitz, 1978*). These findings not only demonstrate noradrenergic modulation of food intake, but also indicate that the NA effect on feeding behavior is versatile and site-specific. Additional evidence that NA is associated with the regulation of the homeostatic system in the ARH came from extracellular recordings of unidentified ARH neurons in rats, which responded to NA application (*Kang et al., 2000*). Furthermore, NA increases the expression of *Agrp* and *Npy* mRNA in the rat ARH in response to glucoprivation (*Fraley and Ritter, 2003*). This effect is abolished if NA terminals in the ARH are eliminated by the injection of saporin-conjugated anti-dopamine-$\beta$-hydroxylase (anti-d$\beta$h), a selective NA immunotoxin. Specifically, the immunotoxin also eliminated a large number of neurons in the ventrolateral medulla (A1/C1 cell groups), which have been identified as mediators of glucoprivic feeding. Together, these findings provide strong evidence that NA modulates eating behavior by changing functional properties of ARH neurons.

Recent work by the Bouret lab provides a plausible hypothesis in regard to the question in which behavioral context the noradrenergic LC projections into the ARH play a key role and what response they facilitate. Based on in vivo electrophysiological recordings in primates, it is hypothesized that the LC plays a key role in mobilizing the required energy to face anticipated physical challenges, such as to obtain rewards (*Bouret and Richmond, 2015*; *Varazzani et al., 2015*). In this model, the LC activity is directly related to the energetic need that is required to master the challenge. NA release in the ARH might then promote compensatory energy intake.

On the cellular level NA action can be mediated by a variety of ARs that can inhibit or excite neurons. NA and adrenalin activate three major classes of G protein-coupled receptors (GPCRs): $\alpha_1$-, $\alpha_2$- and $\beta$-ARs (*Hein, 2006*). These receptor types can be further subdivided in a number of different isoforms. The intracellular signaling cascades of ARs are well investigated. While excitatory $\alpha_1$-ARs are coupled to Gq/11 and mediate an increase in intracellular $Ca^{2+}$ levels and closure of G protein-coupled inwardly rectifying potassium channels (GIRKs), inhibitory $\alpha_2$-ARs couple to Gi/o leading to inhibition of voltage-gated $Ca^{2+}$ channels (VGCCs) and opening of GIRKs. $\beta$-ARs mediate excitation via activation of Gs proteins and subsequent elevation of internal $Ca^{2+}$ concentrations and opening of VGCCs (*Marzo et al., 2009*).

In this study, we assessed if NA can directly modulate orexigenic NPY/AgRP and anorexigenic POMC neurons, which are key components of the control circuits in ARH that regulate food intake and energy homeostasis. In electrophysiological recordings, NA had clear opposite effects on these neurons, increasing the activity of NPY/AgRP neurons and inhibiting POMC neurons. Cell type-specific transcriptomics and pharmacological experiments revealed that the activation of NPY/AgRP neurons is predominantly mediated by $\alpha_{1A}$-ARs, while POMC neurons are inhibited via $\alpha_{2A}$-ARs. Collectively, our data indicate a strong orexigenic influence of NA on the ARH circuitry that controls energy balance.

## Results

Here we asked, if and how NA directly modulates the activity of NPY/AgRP and POMC expressing neurons of the ARH. This question was addressed in three steps. First, electrophysiological

recordings were performed to analyze the modulatory effect of NA on these neurons. Second, we used cell type-specific transcriptomics to build an inventory of adrenergic receptors that are expressed in NPY/AgRP and POMC neurons and potentially could mediate the modulation of these neurons. Third, pharmacological tools were used to test which adrenergic receptors are functionally expressed and how they contribute to the net modulatory effect of NA.

## Noradrenalin modulates electrophysiological activity differentially in NPY/AgRP and POMC neurons

In the first set of experiments, we probed if NA modulates the electrophysiological activity of NPY/AgRP and POMC neurons. To address this question directly, we analyzed identified NPY/AgRP-GFP and POMC-GFP neurons of the ARH in acute brain slices of adult mice. Recordings were performed in the perforated patch clamp configuration to minimize effects on intracellular signaling pathways.

As described previously, most NPY/AgRP and POMC neurons were spontaneously active under control conditions (NPY/AgRP neurons: (*Baver et al., 2014*; *Steculorum et al., 2015*; *Tsaousidou et al., 2014*; *Wei et al., 2015*); POMC neurons: (*Ernst et al., 2009*; *Koch et al., 2015*; *Newton et al., 2013*; *Plum et al., 2006*; *Zhang and van den Pol, 2013*). Bath-applied 10 µM NA had clear opposite effects on these functionally antagonistic neuron types. NA depolarized orexigenic NPY/AgRP neurons and increased their action potential frequency from 2.0 ± 0.9 Hz to 5.0 ± 1.0 Hz (*Figure 1A,B*; NPY/AgRP neurons, Wilcoxon matched pairs signed rank test, n = 8, p=0.0012). In contrast, NA clearly hyperpolarized anorexigenic POMC neurons and drastically reduced or even abolished spontaneous generation of action potentials from 3.5 ± 1.1 to 0.7 ± 0.7 Hz (*Figure 1A,B*; POMC neurons, Wilcoxon matched pairs signed rank test, n = 8, p=0.0156). In both neuron types, the NA effect had a rapid onset and was readily reversible. To better define and understand the cause-effect relationship of the NA modulation, we measured concentration-response curves for both neuron types.

## NPY/AgRP neurons, NA concentration-response relation

NA depolarized the membrane potential and increased the action potential frequency starting at a concentration of 100 nM (*Figure 2A*). The maximal responses were observed at 10 µM where NA increased the average firing frequency from 2.2 ± 0.8 Hz to 4.7 ± 0.8 Hz (*Figure 2B*; paired t-test, n = 8, p<0.0001) and depolarized the membrane by 5.1 ± 1.0 mV (*Figure 2C*; paired t-test, n = 8, p=0.0011). The normalized concentration-response relations for the membrane depolarization and increase in firing frequency were well fit with a sigmoidal relation (*Equation 1*). The concentration-response relations had $EC_{50}$s of 1.9 µM (1.1–3.2 µM; n = 8) and 1.5 µM (0.8–2.8 µM; n = 8), respectively. The effective concentration range is in line with previous studies that analyzed NA modulation in other cell types (*Hayar et al., 1997*). To rule out that the observed NA effect is caused by secondary modulation, which is mediated by neurons that are presynaptic to NPY/AgRP neurons, we analyzed the NA effect on NPY/AgRP neurons in which GABAergic and glutamatergic input was pharmacologically blocked. Synaptically isolated NPY/AgRP neurons responded similarly to the application of NA as without synaptic blockers (*Figure 2D,E*), demonstrating that the excitatory NA effect is mediated by cell-intrinsic ARs and the applied GABA and glutamate antagonists do not directly interfere with the AR.

## POMC neurons, NA concentration-response relation

10 µM NA clearly inhibited POMC neurons (*Figure 1A–C*). In neurons with low control action potential frequencies, this often caused an all or nothing effect. Therefore, we used the membrane potential and membrane conductance densities to quantify the NA effect and to establish concentration-response relations.

The NA inhibition started at 100 nM and the maximal inhibitory effect was reached at 10 µM where NA hyperpolarized the membrane potential by 18.6 ± 3.2 mV (*Figure 3B*; paired t-test, n = 8; p=0.0007) and increased the conductance from 647 ± 106 pS to 1546 ± 266 pS (*Figure 3C*; paired t-test, n = 8; p=0.0029). The NA effect developed rapidly and was reversible. However, after application of high concentration, the wash out of NA induced a slow, sustained (~10 min) rebound excitation. NA concentrations higher than 10 µM did not further increase the effect.

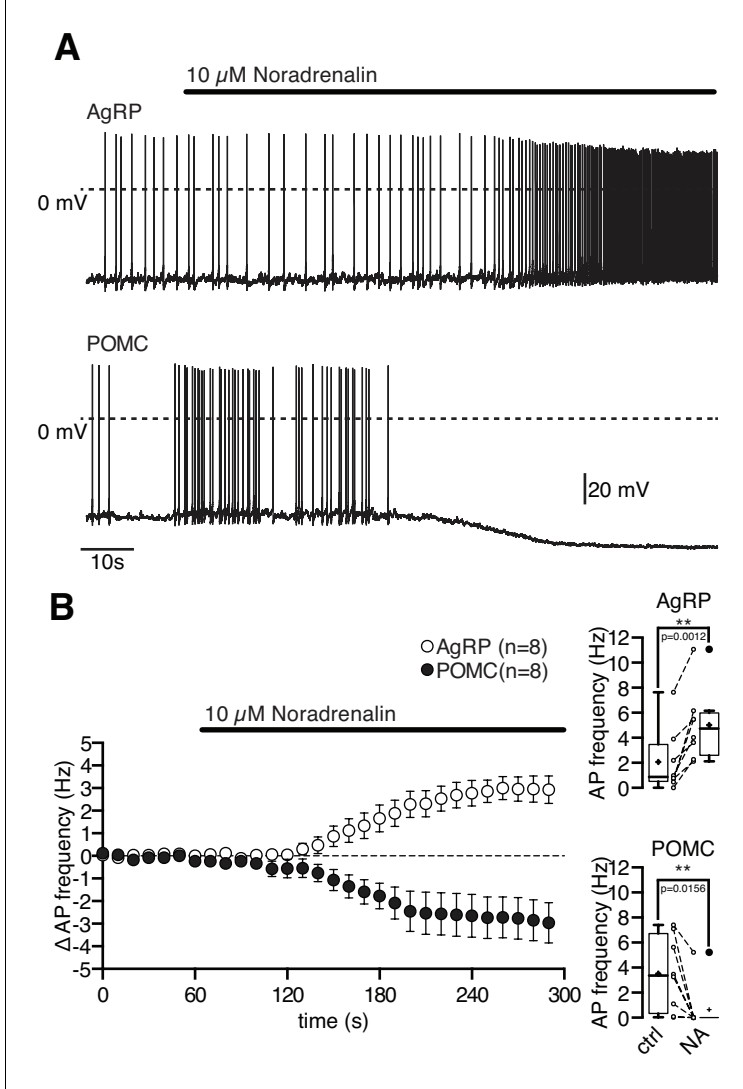

**Figure 1.** Noradrenalin differentially modulates functionally antagonistic NPY/AgRP and POMC neurons. NA (10 μM) excited orexigenic NPY/AgRP neurons and inhibited anorexigenic POMC neurons. Original recordings (**A**) and averaged responses (**B**) of NPY/AgRP (n = 8) and POMC neurons (n = 8) during NA application. The mean response is expressed as change in action potential frequency. The boxplots show the absolute change in action potential frequency for both neuron populations. **p<0.01, Wilcoxon matched pairs signed ranks test.

The normalized concentration-response relations for conductance increase and membrane hyperpolarization were well fit with a sigmoidal relation (*Equation 1*), with $EC_{50}$s of 0.9 μM (0.6–1.5 μM; n = 8) and 1.3 μM (1.0–1.9 μM; n = 8) for the hyperpolarization and the conductance density, respectively. The effective concentration range is in line with previous studies that analyzed NA modulation in other cell types (*Jurgens et al., 2007*).

In experiments in which the GABAergic and glutamatergic input of the POMC neurons was pharmacologically blocked, the NA response was slightly decreased (~16%) compared to recordings without synaptic blockers present, suggesting a small presynaptic contribution by NA responsive neurons, which project on POMC neurons (*Figure 3D,E*). Given the excitatory effect of NA on NPY/AgRP neurons, this effect is likely caused by their mono-directional inhibitory synaptic innervation of POMC neurons.

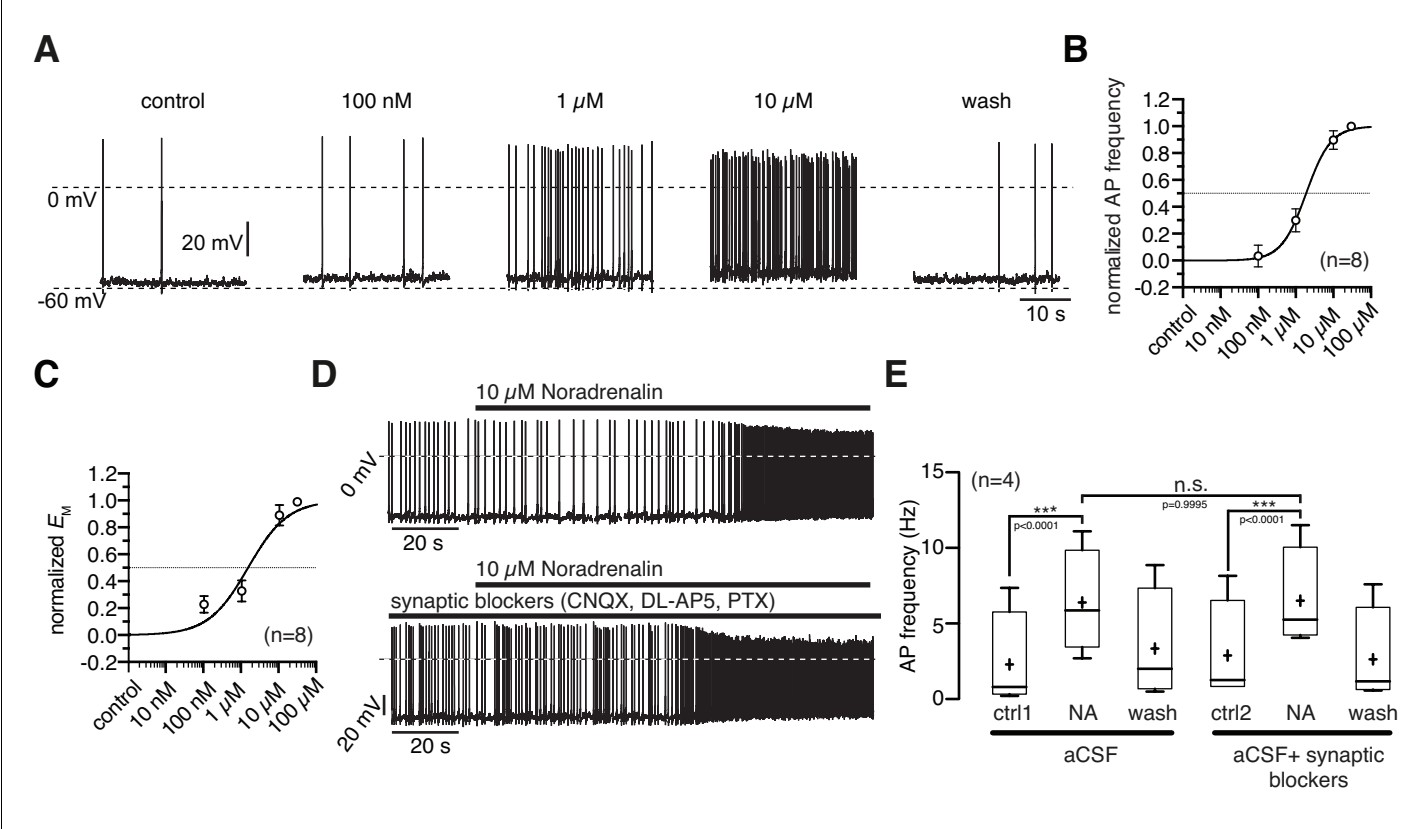

**Figure 2.** Noradrenergic modulation of NPY/AgRP neurons is concentration dependent. (**A**) Recording of a NPY/AgRP neuron demonstrating the effect of increasing NA concentrations. (**B**) and (**C**) *Concentration - response* relations showing the NA effect on action potential frequency (**B**) and membrane potential (**C**). The curves fit to a sigmoidal relation (**Equation 1**). NA had an EC$_{50}$ of 1.9 µM (1.1–3.2 µM; n = 8) for the AP frequency and 1.5 µM (0.8–2.8 µM; n = 8) for the membrane potential, respectively. (**D**) and (**E**) The NA modulation of NPY/AgRP neurons is direct and not dependent on synaptic input. Original recording (**D**) and averaged effect on action potential frequency (**E**) showing that the NA effect on NPY/AgRP neurons is not changed when glutamatergic and GABAergic synaptic input is blocked. Experiments in (**E**) were performed consecutively with the same set of neurons (n = 8). Control 1 and control two refers to the different starting conditions, that is preincubation with or without synaptic blockers. ***p<0.001, one-way ANOVA with post hoc Tukey analysis.

The following figure supplement is available for figure 2:

**Figure supplement 1.** Repeated applications of noradrenaline do not cause desensitization.

## Adrenergic receptors are expressed in NPY/AgRP and POMC neurons of the ARH

To understand how NA modulates NPY/AgRP and POMC neurons at the molecular level, we performed cell type-specific transcriptomics to analyze AR gene expression in these neurons. Using enhanced green fluorescent protein (eGFP)-based fluorescence-activated cell sorting (FACS), we isolated eGFP-expressing NPY/AgRP or POMC neurons from coronal slices, which contained the hypothalamus, of 6-week-old mice (four mice per sample; NPY-GFP: n = 3; POMC-GFP: n = 2). Subsequently, we performed total RNA isolation and whole transcriptome analysis from these neuronal populations. Comparing POMC versus NPY transcriptional profiles yielded 4482 significantly differentially expressed genes (FC≥2, p≤0.01). Within this pool, *Pomc* was enriched 147-fold in the corresponding population, with *Npy* and *Agrp* being higher expressed in NPY-derived samples, but not in a statistically significant way (**Figure 4A**, **supplemental file 1**). NPY/AgRP neurons expressed all subtypes of ARs, except α$_{1D}$- and α$_{2B}$- ARs. POMC neurons only expressed α$_{1A}$- and α$_{2A}$- ARs, albeit to a lower magnitude (**Figure 4A**).

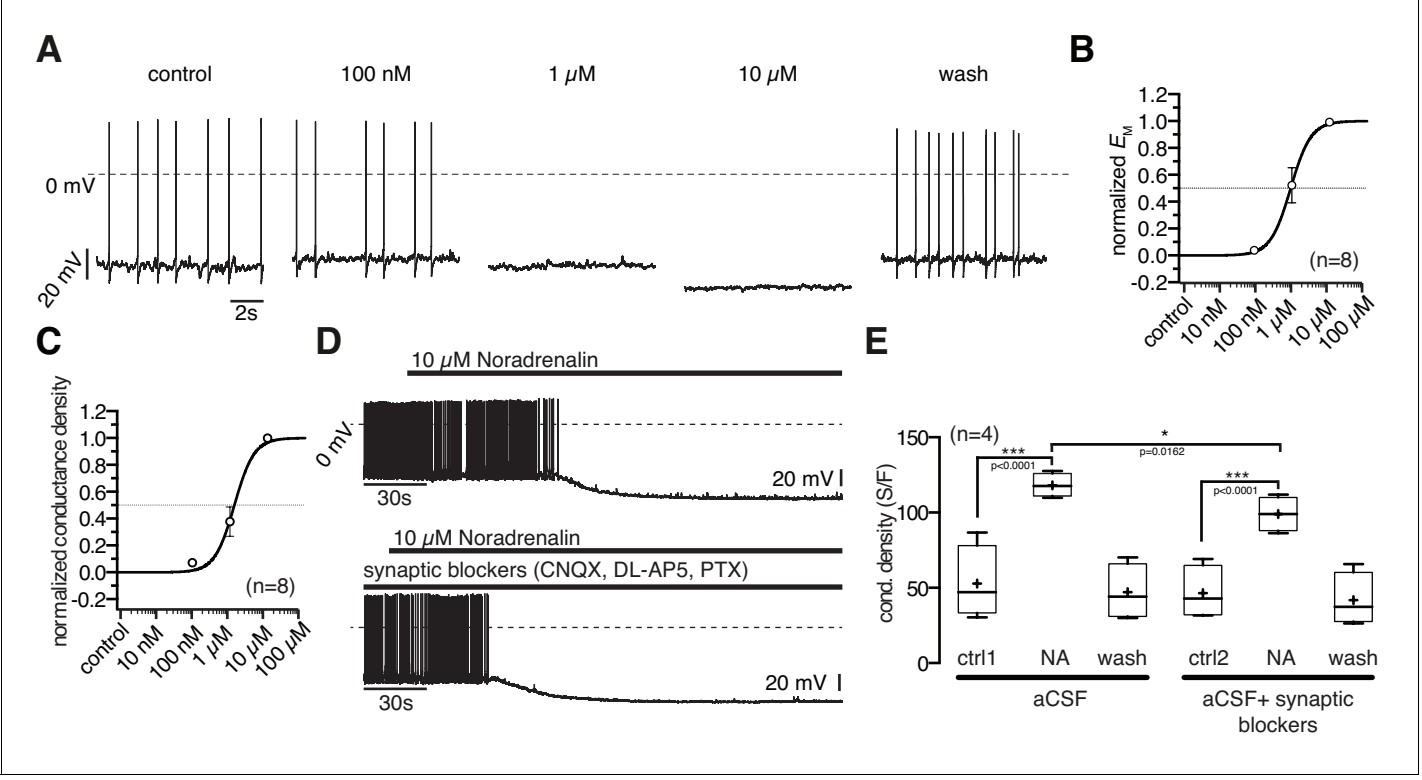

**Figure 3.** Noradrenergic modulation of POMC neurons is concentration dependent. (A) Recording of a POMC neuron demonstrating the effect of increasing NA concentrations. (B) and (C) *Concentration - response* relations showing the NA effect on membrane potential (B) and conductance density (C). The curves are fits to a sigmoidal relation (*Equation 1*). NA had an $EC_{50}$ of 0.9 µM (0.6–1.5 µM; n = 8) for the membrane potential and 1.3 µM (1.0–1.9 µM; n = 8) for the conductance density, respectively. (D) and (E) The NA modulation of POMC neurons is direct and not dependent on synaptic input. Original recording (D) and averaged effect on conductance density (E) showing that the NA effect on POMC neurons is not changed when glutamatergic and GABAergic synaptic input is blocked. Experiments shown in (E) were performed consecutively with the same set of neurons (n = 4). Control 1 and control two refers to the different starting conditions, that is pre-incubation with or without synaptic blockers. **p<0.01; ***p<0.001; one-way ANOVA with post hoc Tukey analysis.

To verify our gene expression data related to the ARs expressed in these neuronal populations, we performed RNA in situ hybridizations, using probes against *Adra1a*, *Adra2a*, *Adrb1* and *Adrb2*. Both NPY/AgRP and POMC neurons were positive for *Adra1a* and *Adra2a* expression, with *Adra1a* being expressed in more NPY/AgRP neurons, compared to POMC neurons (*Figure 4B,C*). *Adra2a* was however expressed to a similar extent between the two populations (*Figure 4C*). *Adrb1* and *Adrb2* were expressed only in NPY/AgRP neurons (*Figure 4—figure supplement 1*). Taken together, our data suggest that ARs are differentially expressed between NPY and POMC neurons within the hypothalamus, possibly mediating any distinct modulatory effects of NA on these populations.

## NA excites NPY/AgRP neurons predominantly via the activation of α₁ₐ-ARs

While NPY/AgRP neurons expressed excitatory α₁ₐ,ᵦ- and β-ARs, and inhibitory α₂ₐ -AR, NA clearly excited all recorded NPY/AgRP neurons. To assess the AR classes that contribute to the excitatory electrophysiological response we used pharmacological tools, which were selective for AR subclasses.

In the first series of experiments α-ARs were blocked by the α₁-AR antagonist prazosine (5 µM) and the α₂-AR antagonist yohimbine (5 µM; *Figure 5A,B*). During the combined application of prazosine and yohimbine the excitatory NA effect was completely blocked in 44% (4 of 9; *Figure 5A*) and significantly (~60%) attenuated in 55% (5 of 9; *Figure 5B*) of the NPY/AgRP neurons. Together,

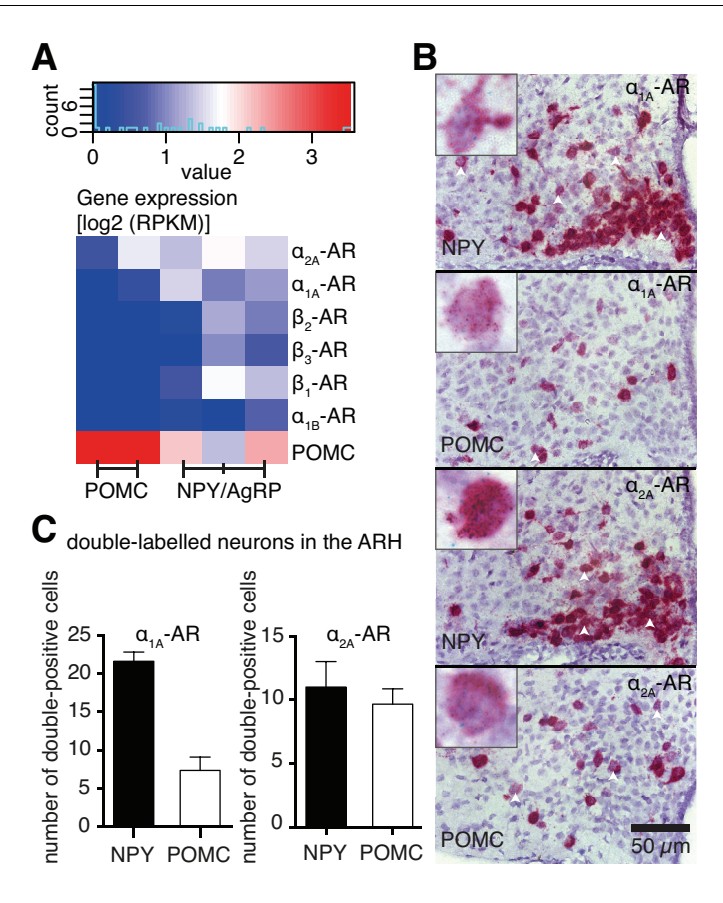

**Figure 4.** Cell type-specific expression of ARs. (**A**) Log expression levels of adrenergic receptor genes in Pomc- and Npy-expressing cell populations. Deeper red colours indicate higher expression levels in the respective cell population. RPKM: reads per kilobase per million mapped reads. (**B**) Images from RNA in situ hybridizations against *Adra1a* (upper panels, green dots) and *Adra2a* (lower panels, green dots) in *Npy*- and *Pomc*- expressing (red) neurons. White arrowheads indicate doubly-labeled cells. Higher magnification indexes are representative of doubly-labeled cells. (**C**) Quantification of double positive cells for *Adra1a* and *Adra2a* in *Npy*- and *Pomc*-expressing neurons.

The following figure supplement is available for figure 4:

**Figure supplement 1.** Images (A,B) and quantification (C) of RNA in situ hybridizations against Adrb1 (upper panels, green dots) and Adrb2 (lower panels, green dots) in Npy- and Pomc- expressing (red) neurons.

these experiments suggest that the excitatory NA effect in NPY/AgRP neurons can be mediated solely by $\alpha_1$-ARs or by the co-activation of $\alpha_1$-ARs and $\beta$-ARs.

Next, we asked which $\alpha_{1A}$-AR subtype(s) mediates the excitatory NA effect. The selective $\alpha_{1A}$-AR agonist A-61603 (1 µM) depolarized the membrane potential and increased the AP frequency in NPY/AgRP neurons similarly as a subsequent NA application (*Figure 6A,B*). Both the A-61603 and the NA effect were fully reversible. In line with these data, we found that the selective $\alpha_{1A}$-AR antagonist WB4101 (100 nM) abolished the excitatory NA response (*Figure 6C,D*). Application of WB4101 resulted in an inhibitory effect, suggesting baseline activation of $\alpha_{1A}$-AR. The subtype-specific $\alpha1_{B,D}$-AR antagonist chloro-ethyl-clonidine (CEC; 100 nM) did not block or alter the NA action (*Figure 6E, F*), suggesting that $\alpha_{1B,D}$-ARs do not contribute to the excitation. Taken together, these data show that NA excites NPY/AgRP neurons via activation of $\alpha_{1A}$-ARs.

Of note, the $\alpha_{1A}$-AR antagonist WB4101 not only blocked the NA excitation (as described above). In the presence of WB4101 NA even had an inhibitory effect on the NPY/AgRP membrane potential

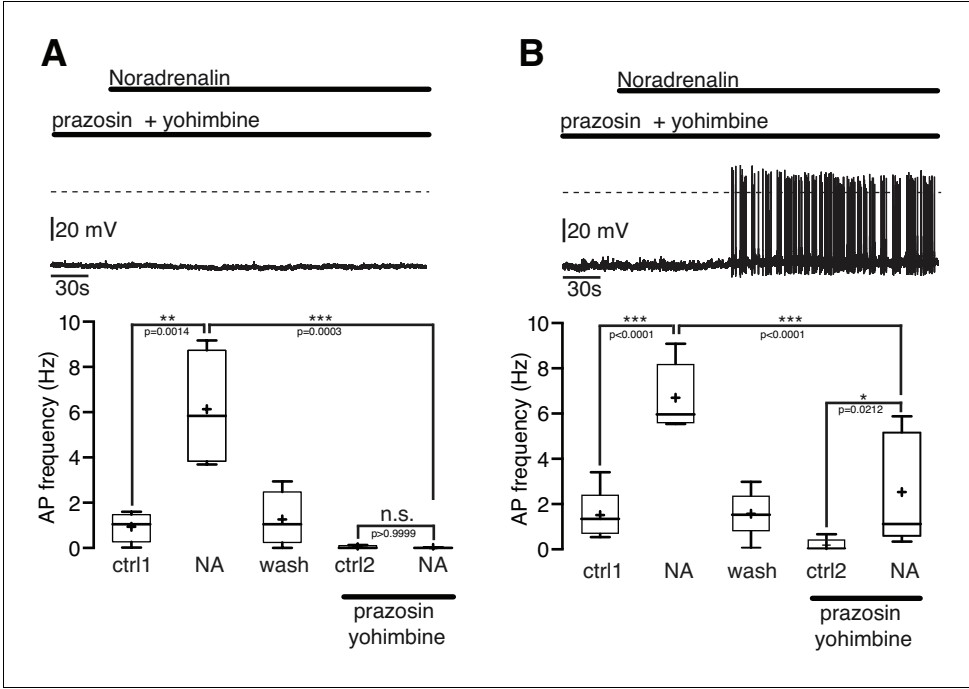

**Figure 5.** The excitatory noradrenergic effect on NPY/AgRP neurons is predominantly mediated by $\alpha_{1A}$-ARs. (**A**) and (**B**) In 44% (4 out of 9) of the NPY/AgRP neurons the NA (10 µM) effect was completely blocked by the $\alpha_1$-AR antagonist prazosine (5 µM) and $\alpha_2$-AR antagonist yohimbine (5 µM) (**A**), while in 56% (5 out of 9) of the NPY/AgRP neurons the NA effect was markedly reduced (~60%) but not completely blocked (**B**). The series of experiments shown in (**A**) and (**B**) were each performed consecutively with the same set of neurons. n values are given in brackets. Control 1 and control two refers to the different starting conditions, that is pre-incubation with different AR antagonist. **p<0.01; ***p<0.001; one-way ANOVA with post hoc Tukey analysis.

(*Figure 6C,D*). A subsequent block of inhibitory $\alpha_{2A}$-ARs by the specific antagonist BRL44408 reversed the inhibition to a small excitation. These data show that the inhibitory $\alpha_{2A}$-ARs are functional in NPY/AgRP neurons and also support the notion that excitatory $\beta$-ARs are activated by NA, as indicated in the first set of experiments.

Collectively, our data indicate that all receptors, which were identified by cell-type specific transcriptomics, are functional in NPY/AgRP neurons. The excitatory NA effect that we observed in virtually all neurons is predominantly mediated by $\alpha_{1A}$-ARs and might be in some neurons co-mediated by $\beta$-ARs. The activation of excitatory AR interferes with the activation of inhibitory $\alpha_{2A}$-ARs.

## NA inhibits POMC neurons via the activation of $\alpha2_A$-ARs

POMC neurons express excitatory $\alpha_{1A}$-ARs and inhibitory $\alpha_{2A}$-ARs, and were markedly inhibited by NA. First, we tested, if the inhibitory NA effect is mediated by $\alpha_{2A}$-AR, as suggested by our transcriptomics data. The $\alpha_{2A}$-AR antagonist BRL 44408 (10 µM) blocked the NA mediated inhibitory effect. Application of 10 µM BRL 44408 alone, did not significantly alter the recorded electrophysiological properties, suggesting no baseline activity of $\alpha_{2A}$-ARs (*Figure 7A*). The specific $\alpha_{2B}$-AR antagonist ARC 239 (1 µM) did not change the inhibitory effect of NA, which suggests that the inhibitory NA action is solely mediated by $\alpha_{2A}$-ARs (*Figure 7B*).

Next, we asked if the excitatory $\alpha_{1A}$-ARs are functional in POMC neurons (*Figure 7C–E*). When the inhibitory $\alpha_2$-ARs were blocked by yohimbine (5 µM), NA excited POMC neurons (*Figure 7D*; paired t-test, n = 10, p=0.0168). This excitation was completely abolished by the $\alpha_1$-AR antagonist prazosine (5 µM), showing the functionality of $\alpha_1$-ARs in POMC neurons (*Figure 7E*; paired t-test, n = 6, p=0.2467).

Taken together, our results show that the NA-induced inhibition of POMC neurons is mediated by $\alpha_{2A}$-AR and can mask the activation of excitatory $\alpha_1$-ARs.

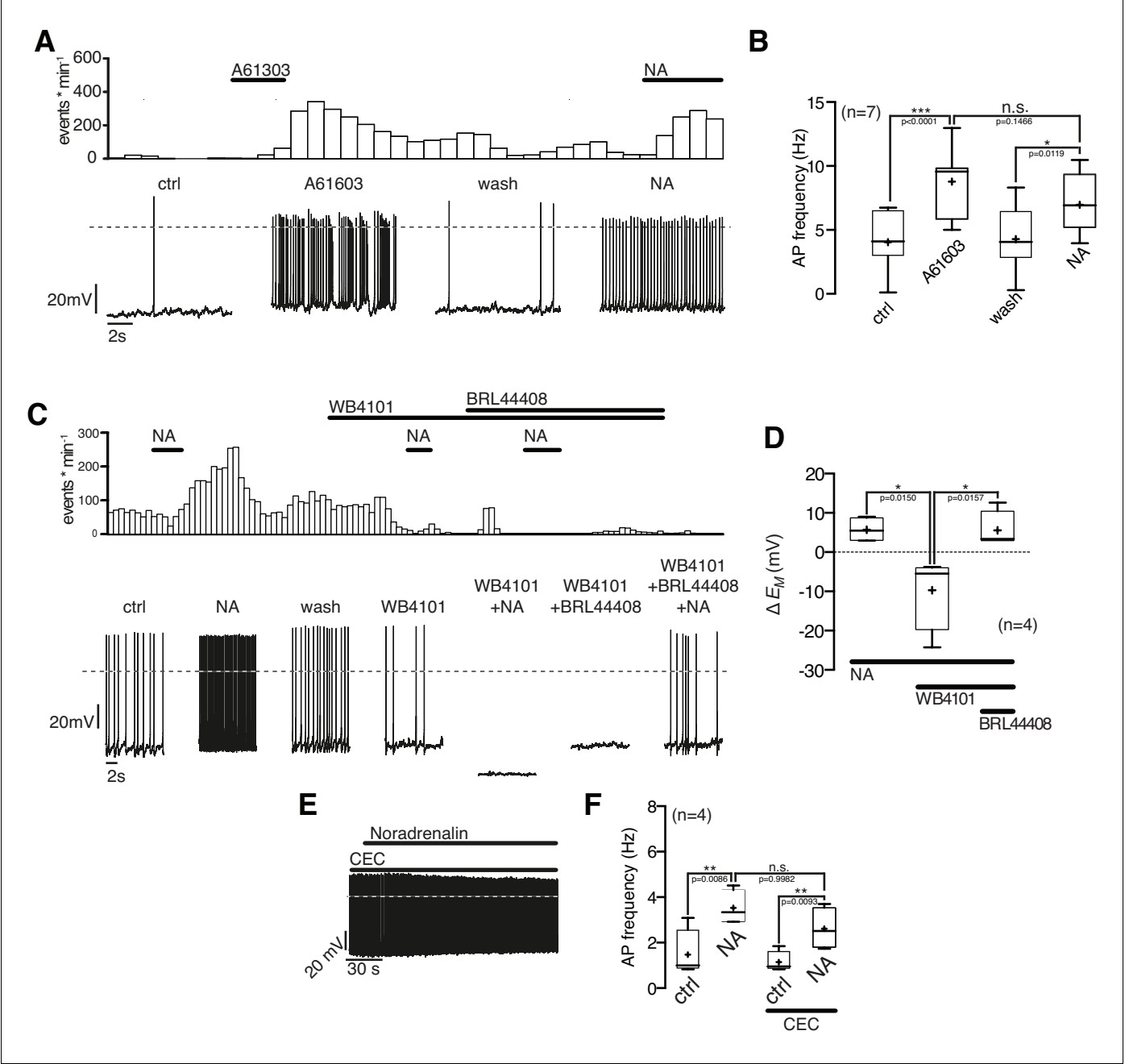

**Figure 6.** NPY/AgRP neurons are excited by $\alpha_{1A}$-AR and inhibited by $\alpha_{2A}$-AR when $\alpha_{1A}$-AR are blocked. (A) Rate histogram with its original recording showing that the $\alpha_{1A}$-AR agonist A61603 (1 µM) had similar excitatory effects on action potential frequency (B) as NA. (C) and (D) The selective $\alpha_{1A}$-AR antagonist WB4101 (100 nM) blocked the excitatory NA effect. In the presence of WB4101 NA had an inhibitory effect on the membrane potential, which was eliminated by the $\alpha_{2A}$-AR antagonist BRL44408 (10 µM). (E) The $\alpha_{1BD}$-AR blocker CEC (100 nM) did not change the NA effect on NPY/AgRP neurons. *p<0.05; **p<0.01; ***p<0.001; one-way ANOVA with posthoc Tukey analysis.

The following figure supplement is available for figure 6:

**Figure supplement 1.** Repeated applications of noradrenaline do not cause desensitization.

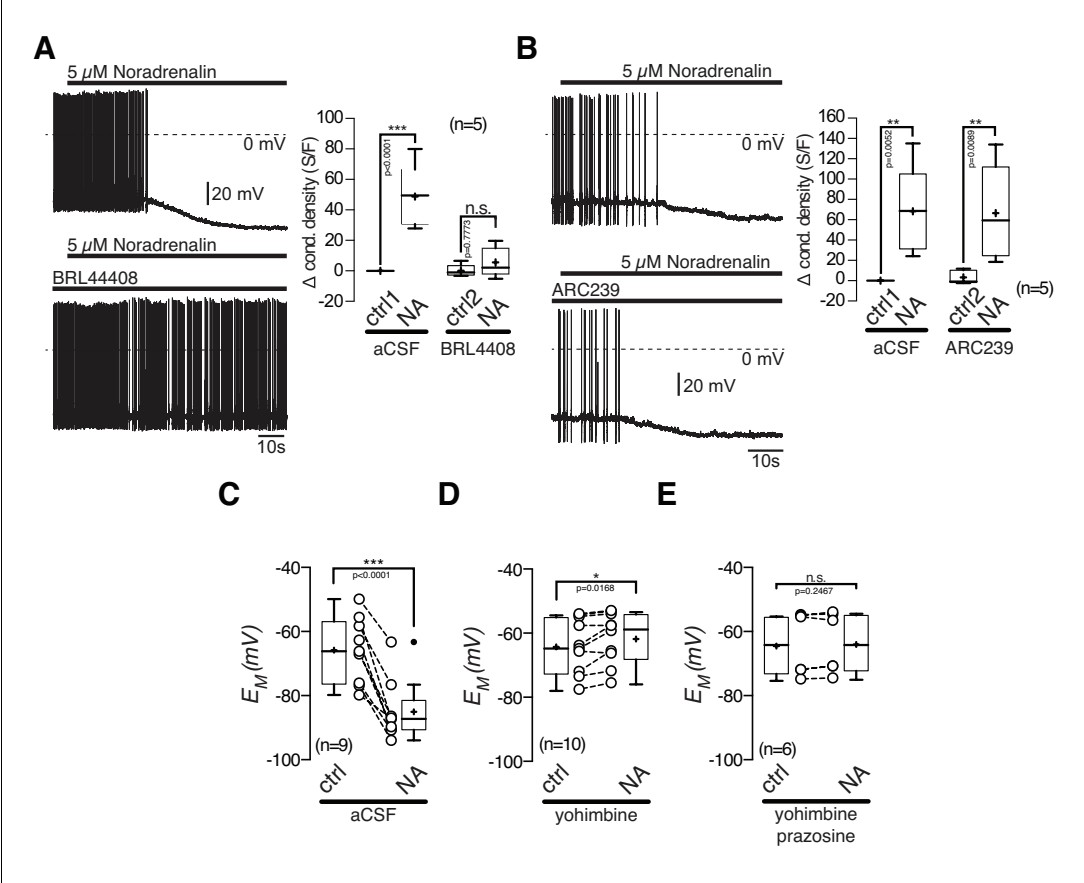

**Figure 7.** The inhibitory noradrenalin effect on POMC neurons is mediated by $\alpha_{2A}$-AR. (**A**) and (**B**) The $\alpha_{2A}$-antagonist BRL 44408 (10 µM) blocked the inhibitory NA (5 µM) effect (**A**), while the $\alpha_{2A}$-antagonist ARC239 (1 µM) did not affect the NA action (**B**). (**C**–**E**) Inhibitory NA effect (**C**). Blocking $\alpha_2$-AR by yohimbine revealed an excitatory NA effect (**D**), which was blocked by the $\alpha_1$-AR antagonist prazosine (**E**). The series of experiments shown in (**A**) and (**B**) were each performed consecutively with the same set of neurons. Control 1 and control two refers to the different starting conditions, that is preincubation with different AR antagonist. n values are given in brackets. *p<0.05; **p<0.01; ***p<0.001. (**A**) and (**B**) one-way ANOVA with post hoc Tukey analysis; (**C**–**E**) paired t-test.

## Discussion

NA neurons of the brainstem give rise to a complex system of fiber tracts innervating most of the forebrain including the hypothalamus and the ARH (*Szabadi, 2013*). Because of the extensive and complex projection pattern, the NA system is associated with a number of important CNS functions including control of wake-sleep-cycles, nociception and memory consolidation (*Berridge and Waterhouse, 2003*; *Samuels and Szabadi, 2008a*, *2008b*). One important role that has been clearly linked to the NA system by a combination of anatomical, pharmacological, lesion and behavioral studies is the modulation of energy homeostasis and food intake (*Wellman, 2005*, *2000*)

Here, we show for the first time that NA directly modulates NPY/AgRP and POMC neurons of the ARH, which are key components of the neuronal circuits that control food intake and energy homeostasis. Interestingly, NA changes the activity of these functionally antagonistic neurons in opposite ways, increasing the activity of the orexigenic NPY/AgRP neurons and decreasing the activity of the anorexigenic POMC neurons. The strong differential modulation, which was reproducible in virtually all recorded neurons, is remarkable, since other important fuel sensing modulators like insulin, leptin, glucose, oleic acid and UDP typically affect only a subpopulation of these neurons (*Jo et al., 2009*; *Parton et al., 2007*; *Steculorum et al., 2015*; *Williams et al., 2010*). On the network and systems level, this clear differential NA modulation of functionally antagonistic neuron types should generate a strong orexigenic drive on the homeostatic circuits of the ARH. This conclusion is consistent with

early behavioral experiments from *Clark et al. (1988)* showing that injection of NA into the third ventricle increased food intake. Furthermore, the authors showed that blockade of $\alpha_2$-AR attenuated the NA-induced feeding behavior, which is in line with our finding that satiety signaling POMC- neurons are inhibited by NA via $\alpha_2$-AR.

Since the modulatory effects of NA were observed in synaptically isolated neurons, the data suggest that NA directly activated intrinsically expressed ARs. Our cell type-specific transcriptomic data showed expression of excitatory and inhibitory receptor subtypes in both NPY/AgRP and POMC neurons, which is in line with recent data from *Henry et al. (2015)*. NPY/AgRP neurons expressed excitatory $\alpha_{1A}$-, $\beta$-ARs and inhibitory $\alpha_{2A}$-ARs, while POMC neurons only express excitatory $\alpha_{1A}$-ARs and inhibitory $\alpha_{2A}$-ARs. Expression of other AR-subtypes was not detected in these neurons.

To unravel which of the expressed AR-subtypes contribute to the cell type-specific physiological effects of NA on NPY/AgRP and POMC neurons, we used pharmacological tools that have been previously shown to be selective for AR subtypes. While in NPY/AgRP the robust excitation is mediated by excitatory $\alpha_{1A}$- and $\beta$-ARs, $\alpha_{2A}$-ARs mediated the strong inhibition in POMC neurons.

The functional consequences of these diverse expression and activation profiles on the systems and behavioral level is not entirely clear and has to be investigated in future studies. Previous studies in other neuron types revealed similar phenomena and suggested that simultaneous expression of excitatory and inhibitory (NA) receptors might derive from developmental effects or localized, differential expression of ARs in different functional compartments.

Simultaneous expression of excitatory and inhibitory adrenergic receptors has been found in neurons of the LC. In juvenile mice, mRNA for excitatory $\alpha_{1A,B,D}$- and inhibitory $\alpha_{2A,B,C}$-ARs has been detected and application of NA caused excitation, while in adult mice NA hyperpolarizes LC neurons. Here, $\alpha_1$-ARs attenuate the NA-induced hyperpolarization (*Osborne et al., 2002*). As in LC neurons, POMC neurons might change their response to NA during development. In the ARH developmental changes of electrophysiological responses to leptin have been described for NPY/AgRP neurons (*Baquero et al., 2014*; *Nilsson et al., 2005*). Simultaneous expression of excitatory and inhibitory ARs has also been found in dentate neurons of the human cerebellum (*Schambra et al., 2005*). Here, $\alpha_{2B}$-ARs are thought to modulate excitatory glutamatergic input, while $\alpha_{1A,B}$-ARs may play a role in timing and computation of inhibitory input. Previous studies in dentate neurons of the cerebellum suggested that $\alpha_1$-ARs are expressed on the cell body while $\alpha_2$-ARs are expressed on the terminals, in order to locally regulate transmitter release (*Milner et al., 2000*).

This raises the question regarding the NA source and under which physiological conditions NA is active to modulate food intake. While the blood brain barrier in adult animals is considered mostly impermeable for catecholamines, there is evidence from early work that catecholamines can cross the blood brain barrier in the hypothalamus (*Weil-Malherbe et al., 1959*). However, the site specific effects of local lesions and local NA injections into the hypothalamus might argue for local, site specific NA release from neurons. In rats, projections from the hindbrain/brainstem and the LC to the ARH have been identified (*Fraley, 2006*; *Fraley and Ritter, 2003*; *Yoon et al., 2013*). Specific deletion of projections from the ventro-lateral medulla (A1/C1 cell group) is sufficient to eliminate an increase in AgRP and NPY mRNA expression in response to glucoprivation induced by 2-DG. Furthermore, homeostatic feeding in response to local hypoglycemia in the hindbrain was decreased (*Fraley and Ritter, 2003*). In this context, recent work in primates from the Bouret group suggested that the LC plays a key role in mobilizing the required energy in order to face anticipated physical challenges, such as obtaining rewards (*Bouret and Richmond, 2015*; *Varazzani et al., 2015*). In this model, the LC activity is directly related to the energetic need that is required to master the challenge. Consistent with this hypothesis, our data suggest that NA release into the ARH upon LC activation generates an orexigenic drive to compensate for the required energy. In line with this hypothesis, previous studies revealed that NA injections into the hypothalamus elicit food intake (*Booth, 1967*; *Leibowitz, 1978*).

Together, these behavioral experiments are strong evidence for the orexigenic nature of NA signaling in the control of food intake. In this context, the orexigenic hormones insulin and leptin may modulate NA signaling in the ARH. While insulin selectively decreased $\alpha_2$-AR expression in the ARH and dorsomedial hypothalamus (*Levin et al., 1998*), leptin inhibits NA release in rats (*Brunetti et al., 1999*; *Francis et al., 2004*), Both effects would augment the net anorexigenic signal. In support of the latter relation, obese mice that lack the gene to produce leptin (Lep^ob/ob)

exhibit increased NA levels within the hypothalamus, which further decreases the anorexigenic signal (*Oltmans, 1983*).

Several drugs that were developed to modify food intake, act on the monoaminergic transmitter system in the brain. Since the molecular and cellular targets that mediate the effects of these drugs are often not clearly identified, they often cause strong side effects (*Hainer et al., 2006*). To successfully support obese patients to manage their food intake with minimized side effects, it is important to precisely define the targets of these drugs. Since the current findings reveal cell type-specific pathways that mediate NA modulation in the ARH they might help to better define specific targets for anti-obesity drugs.

## Materials and methods

### Animal care

All animal procedures were conducted in compliance with guidelines approved by local government authorities (Bezirksregierung Köln, Cologne, Germany) and were in accordance with NIH guidelines. Mice were housed at 22–24°C with a 12 hr light/12 hr dark cycle. Animals had access to water and chow *ad libitum*. All experiments have been performed with adult male and female NPY/AgRP-hrGFP (Lowell 2009) or POMC-EGFP (Cowley 2001) mice. The mice for the electrophysiological recordings were 15–20 weeks old. To facilitate tissue dissociation 6 weeks old animals were used for RNA sequencing.

### Neuronal sorting and RNA sequencing

At 6 weeks of age, NPY/AgRP-hrGFP or POMC-EGFP mice were sacrificed by decapitation and the brain area corresponding to the hypothalamus was micro-dissected under a brightfield stereoscope. Each sample contained hypothalami from four mice (males and females) and tissue samples were processed in ice-cold HBSS, until enzymatic treatment. Samples were centrifuged to remove HBSS and were subsequently incubated for 15 min in pre-incubation buffer, at 37°C. Enzyme mix was added and each sample was manually dissociated by gentle pipetting, throughout a total duration of 35 min and according to manufacturer's instructions (Neural tissue dissociation kit, Miltenyi Biotec, Germany). Subsequently, samples were centrifuged at 500 x g, 10 min, 4°C, and the cell pellet was re-suspended in cold PBS, containing 5% fetal calf serum.

NPY/AgRP-hrGFP or POMC-EGFP-expressing neurons were gated and sorted with a blue laser. GFP-positive neurons were sorted directly into total RNA extraction buffer (PicoPure RNA isolation kit, ThermoFisher Scientific, USA). Samples were lysed for 30 min by incubation at 42°C and subsequently frozen to −80°C, to allow for simultaneous processing among replicates. Total RNA was extracted according to manufacturer's instructions (PicoPure RNA isolation kit, ThermoFisher Scientific, USA) and all samples were DNAse-treated (ThermoFisher Scientific, USA). RNA quality was assessed with an Agilent 2100 bioanalyzer and samples with low degradation (RIN >7.5) were further processed for library preparation and sequencing. Due to low amount of input material, pre-amplification using the Ovation RNASeq System V2 was performed. Total RNA was used for first strand cDNA synthesis, using both poly(T) and random primers, followed by second strand synthesis and isothermal strand-displacement amplification. For library preparation, the Illumina Nextera XT DNA sample preparation protocol (Part # 15031942 Rev. C) was used, with 1 ng cDNA input. After validation (Agilent 2200 TapeStation) and quantification (Invitrogen Qubit System) all five transcriptome libraries were pooled. The pool was quantified using the Peqlab KAPA Library Quantification Kit and the Applied Biosystems 7900HT Sequence Detection. One lane and a paired-end read of 2 × 100 bp on the Illumina**HiSeq 2000 sequencer using v3 chemistry resulted in 39-44Mreads/sample (7.9–9.7 Gb) and a ratio of bases above Q30 of 90.2%. RNA-seq data were analyzed using the QuickNGS pipeline, described elsewhere (*Wagle et al., 2015*) (n = 3 for NPY/AgRP- and n = 2 for POMC-expressing neurons).

Statistically significant (p<0.01) altered genes between the two neuronal populations were analyzed with the Ingenuity Pathway Analysis (IPA) software for gene ontology (GO) enrichment, using either the up-regulated or the down-regulated enrichment mode of the core IPA analysis (POMC-neurons were used as controls – down-regulated pool – and NPY-neurons as samples – up-regulated pool).

## Electrophysiology

### Brain slice preparation

The animals were lightly anesthetized with isoflurane (B506; AbbVie Deutschland GmbH and Co KG, Ludwigshafen, Germany) and subsequently decapitated. The brain was rapidly removed and a block of tissue containing the hypothalamus or brainstem was immediately cut out. Coronal slices (270–300 µm) were cut with a vibration microtome (HM-650 V; Thermo Scientific, Walldorf, Germany) under cold (4°C), carbogenated (95% $O_2$ and 5% $CO_2$), glycerol-based modified artificial cerebrospinal fluid (GaCSF; *Ye et al., 2006* ) to enhance the viability of neurons. GaCSF contained (in mM): 250 Glycerol, 2.5 KCl, 2 $MgCl_2$, 2 $CaCl_2$, 1.2 $NaH_2PO_4$, 10 HEPES, 21 $NaHCO_3$, 5 Glucose and was adjusted to pH 7.2 with NaOH resulting in an osmolarity of ~310 mOsm. Brain slices were transferred into carbogenated artificial cerebrospinal fluid (aCSF). aCSF contained (in mM): 125 NaCl, 2.5 KCl, 2 $MgCl_2$, 2 $CaCl_2$, 1.2 $NaH_2PO_4$, 21 $NaHCO_3$, 10 HEPES, and 5 Glucose and was adjusted to pH 7.2 with NaOH resulting in an osmolarity of ~310 mOsm. First, they were kept for 20 min in a 35°C 'recovery bath' and then stored at room temperature (24°C) for at least 30 min prior to recording. For the recordings, slices were transferred to a Sylgard-coated (Dow Corning Corp., Midland, MI, USA) recording chamber (~3 ml volume) and, if not mentioned otherwise, continuously perfused with carbogenated aCSF at a flow rate of ~2 ml · $min^{-1}$. Recordings in the ARC were made at 24°C using an inline solution heater (SH27B; Warner Instruments, Hamden, CT, USA) operated by a temperature controller (TC-324B; Warner Instruments).

### Patch clamp recordings

Current-clamp recordings of NPY/AgRP- and POMC-expressing neurons in the ARH of 15–20 week old mice were performed in the perforated patch clamp configuration. Neurons were visualized with a fixed stage upright microscope (BX51WI, Olympus, Hamburg, Germany) using 40× and 60× water-immersion objectives (LUMplan FL/N 40×, 0.8 numerical aperture, 2 mm working distance; LUMplan FL/N 60×, 1.0 numerical aperture, 2 mm working distance, Olympus) with infrared differential interference contrast optics (*Dodt and Zieglgänsberger, 1990*) and fluorescence optics. POMC and NPY/AgRP neurons were identified by their anatomical location in the ARH and by their GFP fluorescence that was visualized with an X-Cite 120 illumination system (EXFO Photonic Solutions, Ontario, Canada) in combination with a Chroma 41001 filter set (EX: HQ480/40x, BS: Q505LP, EM: HQ535/50m, Chroma, Rockingham, VT, USA). Electrodes with tip resistances between 4 and 6 MΩ were fashioned from borosilicate glass (0.86 mm inner diameter; 1.5 mm outer diameter; GB150-8P; Science Products) with a vertical pipette puller (PP-830; Narishige, London, UK). All recordings were performed with an EPC10 patch-clamp amplifier (HEKA, Lambrecht, Germany) controlled by the program PatchMaster (version 2.32; HEKA) running under Windows. In parallel, data were recorded using a micro1410 data acquisition interface and Spike 2 (version 7) (both from CED, Cambridge, UK). Data were sampled at 10 kHz and low-pass filtered at 2 kHz with a four-pole Bessel filter. Whole-cell capacitance was determined by using the capacitance compensation (C-slow) of the EPC10. Cell input resistances ($R_M$) were calculated from voltage responses to hyperpolarizing current pulses. The calculated liquid junction potential of 14.6 mV between intracellular and extracellular solution was compensated or subtracted offline (calculated with Patcher's Power Tools plug-in from http://www.mpibpc.mpg.de/groups/neher/index.php?page=software for IGOR Pro 6 [Wavemetrics, Lake Oswego, OR, USA]).

### Perforated patch clamp recordings

Perforated patch experiments were conducted using protocols modified from *Horn and Marty (1988)* and *Akaike and Harata (1994)*. Recordings were performed with ATP and GTP free pipette solution containing (in mM): 128 K-gluconate, 10 KCl, 10 HEPES, 0.1 EGTA, 2 $MgCl_2$ adjusted to pH 7.3 with KOH resulting in an osmolarity of ~300 mOsm. ATP and GTP were omitted from the intracellular solution to prevent uncontrolled permeabilization of the cell membrane (*Lindau and Fernandez, 1986*). The patch pipette was tip filled with internal solution and back filled with internal solution, which contained the ionophore to achieve perforated patch recordings and 0.02% tetraethylrhodamine-dextran (3000 MW, D3308, Invitrogen, Eugene, OR, USA) to monitor the stability of the perforated membrane.

Amphotericin B (A4888; Sigma) was dissolved in dimethyl sulfotrate (DMSO; D8418, Sigma) following the protocols of *Rae et al. (1991)* and *Kyrozis and Reichling (1995)*. The used DMSO concentration (0.1–0.3%) had no obvious effect on the investigated neurons. The ionophore was added to the modified pipette solution shortly before use. The final concentration of amphotericin B was ~200 $\mu$g · ml$^{-1}$. During the perforation process access resistance ($R_a$) was constantly monitored and experiments were started after $R_a$ had reached steady state (~15–20 min) and the action potential amplitude was stable.

## Noradrenalin experiments

Noradrenalin (noradrenaline-bitartrate; I9278, Sigma) was bath-applied at concentrations between 10 nM and 100 µM for ~5 min until the NA effect had reached a steady state. Membrane potentials and AP frequencies were measured as averages from time periods (at least 30 s) when the NA effect had reached steady state. The input resistance of a neuron was determined as the mean from three sets of small hyperpolarizing current pulses (−5 to −20 pA). Conductance densities were determined as the inverse of the input resistance divided by the whole-cell capacitance. Concentration-response relations were normalized to the maximal response and fit with the sigmoidal relation (*Equation 1*)

$$y = \frac{1}{1 + 10^{(LogEC50-X)*slope}} \tag{1}$$

## Drugs

Noradrenaline-bitartrate(10 nM - 100 µM; I9278, Sigma), the $\alpha_1$-AR antagonist prazosine (5 µM; P7791, Sigma), the $\alpha_2$-AR antagonist yohimbine (5 µM; Y3125, Sigma), the $\alpha_{2A}$-AR antagonist BRL 44408 (10 µM, C5776, Sigma), the $\alpha_{2B}$-AR antagonist ARC 239, the $\alpha_{1A}$-AR antagonist WB 4101, the $\alpha_{1C}$, D-AR antagonist CEC (1 nM – 1 µM, Q102, Sigma) and the $\alpha_{1A}$-AR agonist A61603 (100 nM; A5861, Sigma) were added to the normal aCSF.

To reduce glutamatergic and GABAergic synaptic input to the recorded neurons, 10 µM CNQX (6-cyano-7-nitroquinoxaline-2,3-dione, C127, Sigma-Aldrich), 50 µM DL-AP5 (DL-2-amino-5-phosphonopentanoic acid, BN0086, Biotrend), and 100 µM PTX (picrotoxin, P1675, Sigma Aldrich) was added to the extracellular saline.

## RNA in situ hybridizations

In situ hybridizations were performed using the RNAscope 2-plex detection chromogenic kit, according to manufacturer's instructions (Advanced Cell Diagnostics, USA). Briefly, wild-type C57BL/6 mice were intra-cardially perfused with 4% paraformaldehyde, post-fixed overnight in 4% PFA at room temperature and subsequently cryo-protected in 25% sucrose in 1x PBS. Coronal slices containing the hypothalamus were boiled at 98°C for antigen retrieval, hybridized with gene-specific probe sets against *Adra1a*, *Adra2a*, *Adrb1* or *Adrb2* and *Npy* or *Pomc*, to indicate the corresponding neuronal populations. Staining was evaluated using a brightfield Leica DM5500 B microscope. Per staining, 3 samples with three sections each were used to manually quantify double-positive cells. Per NPY- or POMC- positive cell, only cells containing at least two green (gene-specific) dots were considered as double-positive.

## Data analysis

Data analysis was performed with Spike2 (version 7; Cambridge Electronic Design Ltd., Cambridge, UK), Igor Pro 6 (Wavemetrics, Portland, OR, USA) and Graphpad Prism (version 5.0b; Graphpad Software Inc., La Jolla, CA, USA). If not stated otherwise, all calculated values are expressed as means ± SEM (standard error of the mean). EC$_{50}$ values are expressed as mean followed by their 95% confidence interval (CI) values. The '+' signs in the box plots show the mean, the horizontal line the median of the data. The whiskers were calculated according to the 'Tukey' method. For comparisons of dependent and independent data paired and unpaired t-tests were used, respectively. For multiple comparisons ANOVA with Tukey post hoc analysis was performed. A significance level of 0.05 was accepted for all tests. Significance levels were: *p<0.05, **p<0.01, ***p<0.001. In the figures n values are given in brackets. Exact p-values are reported if p>0.0001.

## Acknowledgements

The authors wish to thank Ali Abdallah, Christoph Goettlinger, Eva Tsaousidou, and Helmut Wratil for excellent technical assistance.

## Additional information

### Funding

| Funder | Grant reference number | Author |
| --- | --- | --- |
| Deutsche Forschungsge-meinschaft | TR-SFB 134/TP A03 | Peter Kloppenburg |
| Excellence Cluster on Cellular Stress Responses in Aging As-sociated Diseases | | Peter Kloppenburg |
| Max-Planck-Gesellschaft | | Jens Brüning |
| EMBO | Marie Curie Long-Term Fellowship | Ismene Karakasilioti |

The funders had no role in study design, data collection and interpretation, or the decision to submit the work for publication.

### Author contributions

LP, Conceptualization, Data curation, Formal analysis, Investigation, Writing—original draft, Writing—review and editing; IK, Formal analysis, Investigation, Writing—review and editing; JA, PF, Formal analysis, Investigation; JB, Resources, Formal analysis, Supervision, Funding acquisition, Writing—review and editing; PK, Conceptualization, Resources, Formal analysis, Supervision, Funding acquisition, Validation, Writing—original draft, Project administration, Writing—review and editing

### Author ORCIDs

Lars Paeger, http://orcid.org/0000-0001-8716-3483
Peter Kloppenburg, http://orcid.org/0000-0002-4554-404X

## Additional files

### Supplementary files

• Supplementary file 1. Significant gene expression differences between POMC- and NPY-expressing neurons. FC, fold difference. p, p-value.

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
