## [Decision Letter]

Thank you for submitting your article "Antagonistic modulation of NPY/AgRP and POMC neurons in the arcuate nucleus by noradrenalin" for consideration by *eLife*. Your article has been reviewed by two peer reviewers, and the evaluation has been overseen by a Reviewing Editor and Richard Aldrich as the Senior Editor. The reviewers have opted to remain anonymous.

The reviewers have discussed the reviews with one another and the Reviewing Editor has drafted this decision to help you prepare a revised submission

Summary:

In this submission, Paeger et al., elucidate the effects of exogenously applied NA on the electrical output of energy-balance-regulating NPY and neurons in the arcuate nucleus of the hypothalamus. They determine that NA activates NPY neurons but inhibits POMC neurons, and go on to elucidate the receptors involved by a combination of molecular biology and functional pharmacology combined with electrophysiology. The authors believe this activity to be dependent upon α1A – and β- adrenergic receptors in NPY/AgRP neurons and α2A – adrenergic receptors in POMC neurons. The finding are interesting, especially in the context of recent work by Buret linking NA-releasing neurons to signaling energy need. The experiments are well performed and the data support the conclusions of the authors. However, there are some concerns of the methodology that limit enthusiasm.

Essential revisions:

Figure 1–Figure 2. While the NPY/AgRP cells demonstrate an increased APF in response to NA, a change in membrane potential is not apparent. This appears opposite to the effects in POMC neurons. Does NA alter the RMP of NPY/AgRP neurons? If not, how does one reconcile this with the WB4101 data (Figure 6) suggesting an α2A-AR dependent regulation of RMP in NPY/AgRP neurons?

Figure 4 appears of limited advancement over previous work published by the Sternson group. Also, the authors use POMC gene expression as a control, however why have the authors not examined NPY or AgRP gene expression to validate the cell type-specificity?

Figure 5 and Figure 6 – The authors use various agonists/antagonists to examine the requirement of ARs in the acute activation of NPY/AgRP neurons. While Figure 6 legend is missing, it's indicated in the Results section that the experiments in Figure 6 (similar to Figure 2, Figure 3 and Figure 5) were performed using sequential applications in the same cell. A primary concern with these experiments is whether there is desensitization of the first application prior to the second test. This point is raised given that Figure 6 suggests that such desensitization may exist as the amplitude of the depolarization/frequency of events in response to A61603 followed by NA appear quite different. The authors should demonstrate subsequent application of all agonists is similar in the first and second application.

Figure 6 and Figure 7 – Is it possible that the dosages used are confounding the interpretation of the current results? In particular, in the current study the authors used 1µM A-61603. A-61603 is highly selective for α1A-AR when compared to α1B-AR or α1D-AR by almost 100 fold or more. However, early studies suggested that A-61603 is selective at binding α1A-AR over α2-AR by as little as 10 fold – Ki 8.89nM for α1A-AR and Ki 100nM for α2-AR (Knepper et al., 1995). Due to the dosage used, how can the authors be certain that the effects observed are not dependent upon α2-ARs.

Similarly, WB4101 has been shown to label platelet α-2 adrenergic receptors (Sundaresan et al., 1983) and WB4101 appears to bind to a nonadrenergic low-affinity site in both calf and human brain (Weinreich and Seeman, 1981; Lyon and Randall, 1980).

Also, BRL 44408 is selective for α2A-AR over α2B-AR (Ki = 1.7 nM and 144.5 nM), however the authors used BRL 44408 at 10µM likely eliminating this selectivity. These points warrant further experimentation if the authors are to conclude AR specific effects.

Given the authors focus on the role of NPY/AgRP and POMC neurons in feeding behavior, how might the current results fit with earlier work which suggested that α2-AR blockade attenuates feeding behavior induced by both NPY and epinephrine (Clark et al., 1988).

Why are the resting membrane potentials of POMC neurons so hyperpolarized in Figure 7? Results, subsection “Noradrenalin modulates electrophysiological activity differentially in NPY/AgRP and POMC neurons” suggests that POMC neurons were spontaneously active under control conditions. Were these neurons spontaneously active? A search of the literature (including work from the same authors e.g. L. Plum et al., 2006, AJ Newton et al., 2013) reports RMP of POMC neurons from ~-40mV to ~-60mV. RMPs of -70mV or -80mV would appear to exceed this range and these cells appear quite different from those reported in Figure 3. Could the data be susceptible to cell selection biases?

---

## [Author Response]

*Essential revisions:*

*Figure 1–Figure 2. While the NPY/AgRP cells demonstrate an increased APF in response to NA, a change in membrane potential is not apparent. This appears opposite to the effects in POMC neurons. Does NA alter the RMP of NPY/AgRP neurons? If not, how does one reconcile this with the WB4101 data (Figure 6) suggesting an α2A-AR dependent regulation of RMP in NPY/AgRP neurons?*

The reviewers raise an important point. The NPY/AgRP neurons increased APF *and* depolarized in response to NA, which was stated in Results, subsection “NPY/AgRP neurons, NA concentration-response relation” (5.1 ± 1.0 mV, paired t-test, n = 8, p = 0.0011). Compared to the hyperpolarization of the POMC neurons the depolarization of the NPY/AgRP neurons is smaller. Given the scaling in Figure 1 and Figure 2 (where we show the full amplitude of the AP), this depolarization is less obvious. In the revised Figure 2 we have added a line, which indicates the control membrane potential to better visualize the NA induced depolarization. Note that Figure 1 and Figure 2 only show the initial phase of the NA response.

*Figure 4 appears of limited advancement over previous work published by the Sternson group. Also, the authors use POMC gene expression as a control, however why have the authors not examined NPY or AgRP gene expression to validate the cell type-specificity?*

We agree with the reviewers – in hindsight. The gene expression analysis was performed because our initial electrophysiological experiments revealed a clear differential modulation: excitation in NPY/AgRP and inhibition in POMC neurons. Accordingly, we expected clear differential expression of excitatory and inhibitory ARs, which did not match the data by the Sternson lab. To clarify this discrepancy, we repeated the gene expression analysis, which confirmed the data from the Sternson lab, and unequivocally revealed that ARs are differentially expressed between NPY and POMC neurons.

*Figure 5 and Figure 6 – The authors use various agonists/antagonists to examine the requirement of ARs in the acute activation of NPY/AgRP neurons. While Figure 6 legend is missing, it's indicated in the Results section that the experiments in Figure 6 (similar to Figure 2, Figure 3 and Figure 5) were performed using sequential applications in the same cell. A primary concern with these experiments is whether there is desensitization of the first application prior to the second test. This point is raised given that Figure 6 suggests that such desensitization may exist as the amplitude of the depolarization/frequency of events in response to A61603 followed by NA appear quite different. The authors should demonstrate subsequent application of all agonists is similar in the first and second application.*

General response: We apologize for the errors in Figure 6. We now provide the right legend for Figure 6. In addition we have corrected an error in Figure 6: the bar indicating the NA application is shifted to the correct position.

We absolutely agree with the reviewers that repeated drug applications potentially can cause desensitization. However, note that this experimental scheme allowed us to confirm that each recorded cell was responsive to NA in the first place. We consider this type of “intrinsic control” as a real experimental advantage. In our opinion, this did not compromise our conclusions (see below).

Specific response: As suggested by the reviewer we tested if desensitization for NA and A61603 occurs. Under our experimental conditions, the effect of both substances is reversible and did not cause desensitization (NA: Figure 2—figure supplement 1 and A61603: Figure 6—figure supplement 1).

NA: The effect of NA is reversible and we did not observe desensitization using sequential application. In our opinion this supports the validity of the conclusions that we have drawn from the data of Figure 2, Figure 3 and Figure 5. In Figure 2 NA induced the same response in control conditions and in synaptic blockers, suggesting that the excitatory NA effect is fully mediated by cell-intrinsic ARs (Results, subsection “NPY/AgRP neurons, NA concentration-response relation”). These data alone show the lack of desensitization. In Figure 3 NA induced a smaller response during the NA application in synaptic blockers. Since we have shown that the reduced response is not caused by desensitization (Figure 2—figure supplement 1), we conclude that the NA effect is intrinsic and in (a small) part mediated by presynaptic effects (Results, subsection “POMC neurons, NA concentration-response relation”). Since the same line of arguments is valid for the data shown in Figure 5, we conclude that the complete (Figure 5) and the partial block (Figure 5) of the NA response is caused by the AR blockers (Results, subsection “NA excites NPY/AgRP neurons predominantly via the activation of α_1A_-Ars”) and not by desensitization.

A61603: The effect of A61603 is reversible and we did not observe desensitization using sequential application. Figure 6 showed that the α_1A_-AR agonist A61603 can reproduce the excitatory NA effect on the APF in NPY/AgRP-neurons. The main purpose of the following NA applications was to confirm that the investigated neurons were responsive to NA in the first place – which they were. Between the A61603- and the NA response was no significant difference (Figure 6).

Nevertheless, the quantification showed that the NA response tended to be slightly smaller than the A61603 response. This result, however, has to be seen in the context of the subsequent experiments (Figure 6). The analysis on a finer scale revealed that NA not only activated excitatory α_1_-AR, but also inhibitory α_2_-AR. Together this caused a smaller net excitation than A61603 alone. This issue is addressed in more detail next (and in Results, subsection “NA excites NPY/AgRP neurons predominantly via the activation of α_1A_-Ars”).

*Figure 6 and Figure 7 – Is it possible that the dosages used are confounding the interpretation of the current results? In particular, in the current study the authors used 1µM A-61603. A-61603 is highly selective for α1A-AR when compared to α1B-AR or α1D-AR by almost 100 fold or more. However, early studies suggested that A-61603 is selective at binding α1A-AR over α2-AR by as little as 10 fold – Ki 8.89nM for α1A-AR and Ki 100nM for α2-AR (Knepper et al., 1995). Due to the dosage used, how can the authors be certain that the effects observed are not dependent upon α2-ARs.*

We agree with the reviewers that the drug concentrations used in this type of experiments can play an important role for interpreting the data. In this context, it is important to note that it is not straightforward “to transfer” Ki values between preparations. Therefore, our experiments and conclusions are typically based on the application of more than one drug, e.g., the application of agonists and antagonists.

In case of the NPY/AgRP neurons we observed that the net NA effect was excitatory. However, as already mentioned above, a more detailed analysis revealed that theses neurons are excited by α_1A_-AR and inhibited by α_2A_-AR when α-_1A_-AR are blocked. This conclusion is based on the following reasoning: The α_1A_-AR A61603 agonist had a similar excitatory effect as NA, while the α_1A_-AR antagonist WB4101 blocked the excitatory NA effect. Interestingly, in the presence of WB4101 (when the excitatory action is blocked) NA had an inhibitory effect, which could be blocked by the α_2A_-AR antagonist BRL44408. In summary, the experiments show that NA activates both excitatory α_1_-AR and inhibitory α_2_-AR, causing net excitation. This dual effect of NA on NPY/AgRP neurons is explained in the legend of Figure 6 and in text (Results, subsection “NA excites NPY/AgRP neurons predominantly via the activation of α_1A_-Ars”).

*Similarly, WB4101 has been shown to label platelet α-2 adrenergic receptors (Sundaresan et al., 1983) and WB4101 appears to bind to a nonadrenergic low-affinity site in both calf and human brain (Weinreich and Seeman, 1981; Lyon and Randall, 1980).*

We thank the reviewers for pointing out these references.

*Also, BRL 44408 is selective for α2A-AR over α2B-AR (Ki = 1.7 nM and 144.5 nM), however the authors used BRL 44408 at 10µM likely eliminating this selectivity. These points warrant further experimentation if the authors are to conclude AR specific effects.*

We agree with the reviewers that the experiments with the α_2A_-AR blocker BRL 44408 alone are not sufficient to differentiate between α_2A_-AR or α-_2B_-AR inhibitory action. Therefore, we want to refer to the complete series of experiments (Figure 7). The BRL 44408 experiments (Figure 7) and the yohimbine experiments (Figure 7) strongly suggest that the inhibitory NA effect on POMC-neurons is caused by α_2_-AR. Following the argument of the reviewers, these experiments alone cannot differentiate between α_2A_-AR and α_2B_-AR effects. However, since α_2B_-AR antagonist ARC239 did not block the NA inhibitory NA response, we conclude that the inhibitory NA effect is indeed mediated by α_2A_-AR. In addition, and very importantly, the transcriptome of the POMC- neurons did show α_2A_-AR, but no α_2B_-AR expression (Results, subsection “Αdrenergic receptors are expressed in NPY/AgRP and POMC neurons of the ARH”).

*Given the authors focus on the role of NPY/AgRP and POMC neurons in feeding behavior, how might the current results fit with earlier work which suggested that α2-AR blockade attenuates feeding behavior induced by both NPY and epinephrine (Clark et al., 1988).*

Our results fit very well with the behavioral data form Clark et al., 1988. Here we show that NA markedly inhibits satiety signaling POMC neurons via α_2_-AR. This alone (even independently from the excitatory NA action on the hunger signaling NPY/AgRP neurons) would increase the orexigenic drive from the NPY/AgRP – POMC system and should lead to increased food intake. The α_2_-AR antagonist yohimbine blocks the strong inhibitory effect of NA on the POMC neurons, increasing their anorexigenic effect, which in turn should reduce feeding. We have included this Discussion in the revised manuscript (Discussion section).

Given the state of research at the time, Clark et al., 1988 suggested cellular mechanisms, which focused exclusively on the role of NPY. Since, these mechanisms were formulated relatively vaguely at the time (For example: “Yohimine attenuation of NPY and adrenergic transmitter-evoked feeding behavior may involve α_2_-adrenoceptors located on presynaptic elements, or, as suggested by a number of studies (e.g. [18,19]), postsynaptic elements, or both.”) our results neither support nor strictly contradict their hypotheses.

*Why are the resting membrane potentials of POMC neurons so hyperpolarized in Figure 7? Results, subsection “Noradrenalin modulates electrophysiological activity differentially in NPY/AgRP and POMC neurons” suggests that POMC neurons were spontaneously active under control conditions. Were these neurons spontaneously active? A search of the literature (including work from the same authors e.g. L. Plum et al., 2006, AJ Newton et al., 2013) reports RMP of POMC neurons from ~-40mV to ~-60mV. RMPs of -70mV or -80mV would appear to exceed this range and these cells appear quite different from those reported in Figure 3. Could the data be susceptible to cell selection biases?*

First of all, we want to apologize for not having stated the age of the mice that were used for the electrophysiological recordings. The recordings have been performed on 15 – 20 weeks old animals, which is now clearly stated in ‘Materials and methods, subsection “Animal care” and subsection “Electrophysiology” of the revised manuscript. This is important to consider, since previous work suggests that the POMC neurons hyperpolarize with increasing age (Yang et al., 2012 Neuron 75:425-436; Newton et al., 2013). Yang et al., report a mean RMP of ~-65 mV in 24 weeks old mice (estimated from their Figure 1). The authors do not provide absolute values, but the boxplot shows several neurons that are clearly more hyperpolarized. In Newton et al., we report a mean RMP of -60 mV (SEM: 1.1; SD: 5.2; Range: -68 mV to -50 mV) in 15-week-old mice.

In the current manuscript, the boxplot (Figure 7) shows the whole range of the measured RMP with a mean of 65.3 mV (SEM: 3.4; SD: 10.2). Based on the previous reports this is in a range, which we would expect for this age and it is also consistent with data in Figure 3. The boxplot (Figure 7) also reveals that the mean is influenced by 3 very hyperpolarized neurons, while the other 6 neurons are more depolarized. Since the recordings of the hyperpolarized neurons were of high quality, we had no reason to classify them as outliers. The boxplot also shows that NA affects all neurons in the same way, independently of the control RMP. Thus, the NA effect is not limited to the hyperpolarized neurons or stronger in these neurons.

In Figure 7 we replaced the example, which showed a very hyperpolarized RMP, with an example in which the RMP is close to the mean RMP.